# Using machine learning to construct TOMCAT model and occultation measurement-based stratospheric methane (TCOM-CH4) and nitrous oxide (TCOM-N2O) profile data sets

**Sandip S. Dhomse**[1,2] **and Martyn P. Chipperfield**[1,2]

[1]School of Earth and Environment, University of Leeds, Leeds, UK
[2]National Centre for Earth Observation, University of Leeds, Leeds, UK

**Correspondence:** Sandip S. Dhomse (s.s.dhomse@leeds.ac.uk)

**Abstract.** Monitoring the atmospheric concentrations of greenhouse gases (GHGs) is crucial to improve our understanding of their climate impact. However, there are no long-term profile data sets of important GHGs that can be used to gain a better insight into the processes controlling their variations in the atmosphere. In this study, we apply corrections to chemical transport model (CTM) output based on profile measurements from two solar occultation instruments: the HALogen Occultation Experiment (HALOE) and the Atmospheric Chemistry Experiment – Fourier Transform Spectrometer (ACE-FTS). The goal is to construct long-term (1991–2021), gap-free stratospheric profile data sets, hereafter referred to as TCOM, for two important GHGs.

To estimate the corrections that need to be applied to the CTM profiles, we use the extreme gradient boosting (XGBoost) regression model. For methane (TCOM-CH4), we utilize both HALOE and ACE satellite profile measurements from 1992 to 2018 to train the XGBoost model, while profiles from 2019 to 2021 serve as an independent evaluation data set. As there are no nitrous oxide ($N_2O$) profile measurements for earlier years, we derive XGBoost-derived correction terms to construct TCOM-N2O profiles using only ACE-FTS profiles from the 2004–2018 time period, with profiles from 2019–2021 used for the independent evaluation.

Overall, both TCOM-CH4 and TCOM-N2O profiles exhibit excellent agreement with the available satellite-measurement-based data sets. We find that compared to evaluation profiles, biases in TCOM-CH4 and TCOM-N2O are generally less than 10 % and 50 %, respectively, throughout the stratosphere. The daily zonal mean profile data sets, covering altitude (15–60 km) and pressure (300–0.1 hPa) levels, are publicly available via the following links: https://doi.org/10.5281/zenodo.7293740 for TCOM-CH4 (Dhomse, 2022a) and https://doi.org/10.5281/zenodo.7386001 for TCOM-N2O (Dhomse, 2022b).

## 1 Introduction

After carbon dioxide ($CO_2$), methane ($CH_4$) and nitrous oxide ($N_2O$) are currently the two most important anthropogenically emitted greenhouse gases (GHGs), and their concentrations in the atmosphere are increasing at substantial rates (e.g. Meinshausen et al., 2020). The primary natural sources of $CH_4$ are wetlands, the decay of organic waste and livestock, whereas anthropogenic sources include landfills and the production and transport of coal, natural gas and oil (e.g. Saunois et al., 2016; Lan et al., 2021). The primary emission sources for $N_2O$ are agricultural practices, industrial activities, the combustion of fossil fuels and the treatment of solid/liquid waste (e.g. Tian et al., 2020). Importantly, as measured by the global warming potential (GWP), $CH_4$ is about 25 times and $N_2O$ is about 300 times more potent as GHGs compared to $CO_2$.

The lifetime of $CH_4$ in the troposphere is about 9 years (e.g. Lelieveld et al., 1998), and it is primarily removed through oxidation by OH. However, in the stratosphere, $CH_4$ destruction is much slower; hence, its local lifetime increases to about 150 years (Chipperfield et al., 2013). $CH_4$ oxidation is also an important source of water vapour in the stratosphere, which plays a key role in ozone chemistry via $HO_x$ cycles, so it also influences the radiative balance in the middle stratosphere. The primary atmospheric sink for $N_2O$ is photolysis (producing $N_2 + O$) in the stratosphere/mesosphere, so it is also a long-lived species (lifetime about 120 years (Chipperfield et al., 2013)). A secondary sink for $N_2O$ is the reaction with $O(^1D)$ to produce NO, which plays an important role in the middle-atmosphere $O_3$ budget via the $NO_x$ cycle. An important aspect is that increases in both OH and NO can also have positive impacts on ozone, especially in the lower stratosphere, as they help to convert reactive species to long-lived reservoir species. For example, $OH + NO_2 (+ M)$ leads to $HNO_3$ formation, while $CH_4 + Cl$ leads to HCl formation, reducing concentrations of reactive $NO_2$ and Cl. Additionally, as both $CH_4$ and $N_2O$ are long-lived in the stratosphere, monitoring their concentrations also helps us to understand changes in stratospheric chemistry and dynamics.

However, despite their importance, there are only a few satellite instruments that provide global stratospheric profiles of $CH_4$ or $N_2O$. Relatively long-term and high-quality data records are available from two solar occultation instruments, the HALogen Occultation Experiment (HALOE) and the Atmospheric Chemistry Experiment – Fourier Transform Spectrometer (ACE-FTS), and from limb sounding instruments such as the Michelson Interferometer for Passive Atmospheric Sounding (MIPAS) and the Microwave Limb Sounder (MLS). However, these instruments have different spatial and temporal coverages and they use different measurement techniques and retrieval algorithms. Hence, merging these satellite data to construct a single long-term data set for a given species is quite challenging.

Therefore, although stratospheric $CH_4$ and $N_2O$ profile data sets were released recently by Hegglin et al. (2021), they did not attempt to merge data from different satellite instruments. Briefly, these data sets were released as part of the Stratospheric and Tropospheric Processes And their Role in Climate (SPARC) Data Initiative and contain monthly mean zonal mean profiles in volume mixing ratio (vmr) units at pressure levels. Data from individual satellite instruments are averaged at 36 latitude bins (2.5° latitudinal resolution) and provided at 26 pressure levels ranging from 300 to 0.1 hPa. SPARC $CH_4$ profile data are constructed using ACE-FTS (2004–2019), HALOE (1991–2005) and MIPAS (2002–2012) measurements (Hegglin et al., 2020). Note that SPARC data use an earlier (v3.6) version of the ACE-FTS data. For $N_2O$, there is no data set for the 1990s, but for later periods, SPARC $N_2O$ data contain monthly mean values from Aura-MLS (based on v4.2), MIPAS (v224), the Sub-

Millimetre Radiometer (SMR, v2.1) and ACE-FTS measurements. Monthly means values are available only if there are more than five valid profiles for a given latitude/altitude range. Monthly mean files are available for individual instruments and there is no merging or adjustment for different data sets.

To our knowledge, until now, no attempt has been made to merge satellite data records to construct long-term stratospheric $CH_4$ and $N_2O$ profile data sets. Here, we do this by constructing correction terms for the stratospheric $CH_4$ and $N_2O$ profiles from a chemical transport model by analysing the difference between the model and available satellite observations. Then, the correction terms (i.e. the differences that are needed to adjust the TOMCAT $CH_4/N_2O$ profiles) are calculated for all the model grid points to construct a long-term, gap-free stratospheric profile data set. Details of the satellite data and model setup used here are given in Sects. 2 and 3, respectively. The methodology used to estimate the correction terms is described in Sect. 4. An evaluation of the newly constructed data sets for $CH_4$ and $N_2O$ is presented in Sect. 5. Details of data availability are given in Sect. 6, followed by a summary and the conclusions in Sect. 7.

## 2   Satellite data and model setup

Given that $CH_4$ and $N_2O$ are potent GHGs and primary sources of stratospheric water vapour and $NO_x$, stratospheric measurements of $CH_4$ and $N_2O$ gained scientific attention even before the discovery of the Antarctic ozone hole (Farman et al., 1985). Initial measurements were performed by the Stratospheric and Mesospheric Sounder (SAMS) instruments on the Nimbus 7 satellite that was launched in 1978 (Drummond et al., 1980; Jones and Pyle, 1984). Similarly, the Atmospheric Trace Molecule Spectroscopy (ATMOS) instrument (Gunson et al., 1990) provided about 350 profiles during four space shuttle missions (in 1985, 1992, 1993 and 1994). Later, the Improved Stratospheric and Mesospheric Sounder (ISAMS) was able to provide about 2600 profiles per day for about 180 d TS1 between 1991–1992, but retrieval was feasible only for the upper stratospheric/mesospheric altitude range (e.g. Remedios et al., 1996).

A step-change in the number of stratospheric $CH_4$ measurements occurred with the deployment of HALOE on the Upper Atmosphere Research Satellite (UARS) in September 1991, followed by ACE-FTS in August 2003. Both instruments provided about 30 profiles per day (discussed below). Two additional instruments, SCIAMACHY (SCanning Imaging Absorption spectroMeter for Atmospheric CHartographY) and MIPAS on the Envisat satellite platform, also provided useful stratospheric $CH_4$ profiles over the 2003–2012 time period (e.g. Noël et al., 2016, 2018). For $N_2O$, the Cryogenic Limb Array Etalon Spectrometer (CLAES) on the UARS satellite platform provided about 1 year of profile

measurements (October 1991 to July 1992). Later, the Sub-Millimetre Radiometer (SMR) on Odin, launched in 2001 (e.g. Urban et al., 2005), MIPAS, and MLS on the Aura satellite (Waters et al., 2006) also provided very useful $N_2O$ profile measurements. However, to avoid inter-instrument biases, which are likely due to differences in the measurement techniques, we decided here to use only HALOE and ACE-FTS data.

## 2.1 HALOE

HALOE was launched aboard UARS in September 1991 (Russell et al., 1993). The spacecraft was in a 57°-inclined orbit at an altitude of 585 km that allowed for observations from 80° S to 80° N. The HALOE instrument used a combination of broadband radiometry and gas filter correlation techniques to observe several trace gas species in the spectral range of 2.4–10.4 μm (or 963–4140 cm$^{-1}$). HALOE provided about 30 measurements (15 sunrise and 15 sunset) per day with near-global coverage in approximately 1 month. In general, daily measurements are provided at two nearly fixed latitudes (sunrise and sunset) with near-equal longitude spacing. For $CH_4$, the retrieval algorithm uses a 2855–2915 cm$^{-1}$ spectral window (channel 6) and profiles are retrieved for the 15 to 90 km range. The algorithm uses an onion-peeling scheme with 1.5 km thick tangent layer to calculate the transmission using a forward model, thereby achieving about 1.5 km vertical resolution. Here we use HALOE v19 data that is available for the October 1991 to November 2005 time period and is obtained via https://acdisc.gesdisc.eosdis.nasa.gov/data//UARS_HALOE_Level2/ (last access: 6 June 2021).

## 2.2 ACE-FTS

ACE-FTS was launched aboard the SciSat-1 spacecraft in August 2003 (Bernath, 2002). The spacecraft was launched in a drifting orbit at an inclination of 74°, which allows for observations from to 85° S to 85° N. The ACE-FTS instrument has very high spectral resolution (0.02 cm$^{-1}$) and covers the spectral range between 750 and 4400 cm$^{-1}$ (Bernath et al., 2005). Similar to HALOE, ACE-FTS uses the solar occultation technique (30 measurements per day). Global latitude coverage is obtained over a period of 3 months and is almost exactly periodic from year to year. The $CH_4$ profile retrieval uses about 60 microwindows between 1244–3087 cm$^{-1}$, while the $N_2O$ retrieval uses 69 microwindows between 1120–2600 cm$^{-1}$ (Strong et al., 2008). Retrieval is performed at 1 km resolution from about 5 to 70 km (Boone et al., 2020). It is important to note that the ACE retrieval scheme does not use averaging kernels. Rather, it uses a so-called global-analysis-type approach where all data are fitted simultaneously using Levenberg–Marquardt least-squares methods. This means that the vmrs for all the contributing molecules in a given microwindow set are fitted/retrieved simultaneously, which is different to the onion-peeling method adopted for SAGE and HALOE retrievals. Here we use ACE v4.2 data that are obtained via http://www.ace.uwaterloo.ca/data.php (last access: 6 June 2021).

## 3 TOMCAT CTM

As both $CH_4$ and $N_2O$ are long-lived tracers in the stratosphere, their distributions in this region are largely determined by the transport process. Hence, we decided to use profiles simulated by the TOMCAT CTM as it is forced with an up-to-date meteorological reanalysis data set. Briefly, TOMCAT is an off-line three-dimensional CTM that includes a comprehensive stratospheric chemistry scheme, but, in the version used here, with a simple tropospheric chemical scheme (Chipperfield, 2006). This means that concentrations of long-lived ozone-depleting substances (ODSs) and GHGs are prescribed as surface mixing ratio boundary conditions (e.g. WMO, 2018), and these species are assumed to be well mixed throughout the troposphere. For $CH_4$, the model uses observed monthly mean global surface concentrations from the National Oceanic and Atmospheric Administration (NOAA) network. The CTM setup is therefore similar to the control simulations used in our recent studies such as Dhomse et al. (2022) and Li et al. (2022). The model simulation is performed at a 2.8° × 2.8° horizontal resolution with 32 hybrid sigma-pressure levels (surface to about 60 km) and is forced with ERA5 (and ERA5.1) reanalysis meteorology (Hersbach et al., 2020). The effects of time-varying solar flux changes and volcanically enhanced stratospheric aerosol are included by using separate time-varying forcing files (e.g. Dhomse et al., 2015, 2016).

## 4 Methodology

For stratospheric ozone, various methodologies have been adopted to merge different types of data to construct homogenized data sets. They include both simple and complex methodologies for adjusting biases for overlapping time periods (e.g. Hassler et al., 2008, 2018; Arosio et al., 2018) and the use of multivariate linear models (e.g. Randel and Wu, 2007) and data assimilation (Inness et al., 2015). However, we are not aware of any attempt to construct long-term stratospheric $CH_4$ and $N_2O$ profile data sets using different satellite data sets.

Here, our approach is similar to that of Dhomse et al. (2021) for ozone, who used CTM profiles as a transfer function and estimated model-observation biases using machine learning. However, they used observation-based monthly mean zonal mean ozone values from the Stratospheric Water and OzOne Satellite Homogenized (SWOOSH) data set (Davis et al., 2016) rather than individual satellite data products. As there are a number of satellite instruments that provide ozone profile measurements, monthly mean zonal mean

values in merged ozone data sets are considered to be well constrained. However, as noted in Sect. 2, there are very few satellite instruments that provide $CH_4$ profile measurements (mainly two occultation instruments providing 30 profiles per day), so in this study we decided to use individual data points to train a machine learning algorithm. Similarly, for $N_2O$ (among occultation instruments), only ACE-FTS provides a long profile data record, but again it has limited spatial coverage; hence, the calculation of monthly mean zonal mean profiles is subject to sampling errors.

Overall, there are six steps in our approach. First, TOMCAT output fields are sampled for HALOE and ACE measurement collocations. There are about 95 000 HALOE profiles and over 106 000 ACE profiles in the 1991–2021 time period. Second, as ACE profiles are available at 1 km vertical resolution, HALOE profiles are also binned at 1 km vertical resolution and TOMCAT profiles (surface to 60 km) are interpolated to the same grid.

Third, we calculate observation–TOMCAT profile differences for each 1 km grid and satellite measurements are included only if retrieval errors are less than 100 % and retrieved values are greater than zero. Note that we assume that all the measurements with retrieval errors less than 100 % are more or less the absolute truth. Hence, no other uncertainties are considered in the further calculations. Our attempt is to construct profile data that would approximate HALOE/ACE data if the instruments had denser measurements without any temporal gaps. As there are distinct dynamical (and chemical) regimes in the stratosphere in terms of processes controlling the distribution of these two GHGs, we divide global measurements into five latitude bins: Southern Hemisphere (SH) polar (SHpol, 50–90° S), SH mid-latitude (SHmid, 20–70° S), tropical (40° S–40° N), Northern Hemisphere (NH) mid-latitude (NHmid, 20–70° N) and NH polar (NHpol, 50–90° N). A 20° (10° from either side) latitudinal overlap between the bins is allowed to include possible extreme variations in the training data set. Estimated differences for overlapping grids are averaged in order to avoid possible sharp edges near the latitude bin boundaries.

Fourth, we train the extreme gradient boosting (XGBoost) regression model for TOMCAT–observation differences of $CH_4$ or $N_2O$ for each vertical level. This means there is a separate model for each vertical level (46 for 15–60 km) for each of the five latitudinal bins. Briefly, XGBoost is a supervised machine learning algorithm that uses an ensemble of decision trees (e.g. Chen and Guestrin, 2016). XGBoost applies the principle of boosting weak learners using the gradient descent architecture (gradient boosting) with some additional components such as L1 and L2 (Lasso and Ridge) regularization, which helps to prevent over-fitting. There are 13 explanatory variables (or features) in our XGBoost regression model taken from TOMCAT output fields or the ERA5 reanalyses used to force the model. For example, the XG-

Boost regression model for $CH_4$ can be represented as

$$
\begin{aligned}
dCH_4 = {} & \beta_1 CH_4 + \beta_2 O_3 + \beta_3 N_2O + \beta_4 HNO_3 + \beta_5 HCl \\
& + \beta_6 H_2O + \beta_7 HF + \beta_8 NO_2 + \beta_9 ClONO_2 \\
& + \beta_{10} T + \beta_{11} PV + \beta_{12} \Theta + \beta_{13} t + \epsilon,
\end{aligned}
\tag{1}
$$

where $T$ and PV are the temperature and potential vorticity from ERA5 at co-located TOMCAT grid points. Measurement latitude ($\Theta$) and date ($t$) variables are included to represent temporal/spatial variations in the measurements, whereas $\epsilon$ denotes unexplained errors. Each variable $\beta_1$ to $\beta_{13}$ can be considered the contribution coefficient for a given explanatory variable. For $CH_4$, we include an additional (14th) step-function-like term in the XGBoost model that is set to 0 for the HALOE time period and 1 for the ACE-FTS time period. Our approach here is to assume that nearly all differences in the TOMCAT $CH_4$ or $N_2O$ profiles with respect to HALOE and ACE data arise from the incorrect representation of the chemical and dynamical processes in the CTM (including inhomogeneities in ERA5 data that are used to drive TOMCAT transport). Our aim is to find correction terms for the TOMCAT $CH_4$ or $N_2O$ profiles so that they match observational profiles for a particular distribution of model tracers and dynamical setup. Hence, we include nine tracers of varied lifetimes (i.e. $CH_4$, $O_3$, $N_2O$, $HNO_3$, $HCl$, $H_2O$, $HF$, $N_2O$, $ClONO_2$) from TOMCAT. We are aware that some tracers are correlated as all the variables are from a TOMCAT simulation (or forcing meteorology); hence, we use the Lasso (L1) regularization option to remove less important variables in case one or some of them are highly correlated at a particular level. We use the Python package XGBoost (https://xgboost.readthedocs.io/en/stable/python/python_intro.html, last access: 5 January 2023) for the analysis with the following options: n_estimators = 1000, max_depth = 4, alpha = 0.3, learning_rate = 0.1, min_child_weight = 6. As mentioned earlier, profiles prior to 2018 are used to train (70 %) and test (30 %) XGBoost for individual vertical levels. As an additional check, we use the last 3 years (2019–2021) of data points for the evaluation.

Fifth, we sample the daily TOMCAT output at 01:30 and 13:30 UTC equatorial crossing times (daytime and nighttime sampling). The TOMCAT 3D fields are then re-gridded at 1 km vertical resolution before dividing them into five latitude bins (see above). Trained XGBoost regression models are then used to calculate correction terms for all twice-daily 3D output profiles.

Sixth, correction terms for individual model grid points are merged to construct twice-daily (01:30 and 13:30 UTC) 3D (longitude/latitude/height) correction terms. As mentioned above, we use simple averaging for the overlapping grid points to avoid sharp boundaries, followed by simple three-dimensional (latitude–longitude) smoothing using three-point boxcar smoothing. These twice-daily correction terms are then added to the original TOMCAT $CH_4$ and

N$_2$O profiles. Daily mean 3D (longitude/latitude/height) correction terms are calculated by averaging the 01:30 and 13:30 UTC fields, and then zonal means (latitude/height) are calculated to produce daily mean zonal mean TCOM-CH4 and TCOM-N2O profiles.

## 5 Results

As noted in the "Introduction", CH$_4$ and N$_2$O concentrations in the lower stratosphere are largely controlled by dynamical processes. The reanalysis data sets used to drive transport in the CTM can be considered as our best knowledge of the past atmosphere as they attempt to incorporate most of the available high-quality meteorological observations using data assimilation. However, they are prone to issues related to changes in the number and type of observations assimilated in the reanalysis system, which might introduce inhomogeneities into the data sets produced. On the other hand, although chemical models are ideal tools for simulating and understanding past changes in these two greenhouse gases using consistent chemical schemes, they are also prone to deficiencies. For example, some computationally expensive processes (e.g. vertical mixing in the troposphere) are represented by somewhat simplified parameterizations. Additionally, most of the chemical reaction rates (loss rates) calculated in the model scheme can also have large uncertainties. Hence, chemical-transport-model-simulated profiles often show some kind of bias with respect to observational data sets. Similarly, although occultation-technique-based instruments measure atmospheric spectra at relatively high resolution, they also include simplified parameterizations for complex radiative processes (e.g. scattering, the contribution from interfering gases), and so retrieval errors are also sensitive to changes in stratospheric conditions. Hence, here we assume that some of the differences between TOMCAT and observations can be attributed to the distribution of other TOMCAT tracers. We use XGBoost to identify possible interconnection patterns between TOMCAT CH$_4$ or N$_2$O differences and other tracers using available data points so that corrections can be estimated for all model grid points.

Figure 1 shows vertical profiles of estimated variance ($R^2$) and feature (explanatory variable) importances for the SHpol (50–90° S) latitude bin for the XGBoost regression model. Feature importance can be considered as a regression coefficient indicating how much a given variable contributes towards the CH$_4$ or N$_2$O bias-correction prediction. Variance and feature importances for SHmid, tropics, NHmid and NHpol are shown in Figs. S1 to S4 in the Supplement, respectively. For SHpol, XGBoost seems to show excellent performance for both species throughout the stratosphere, with $R^2$ values ranging from 0.6 to 0.8. This also validates our approach of using different long-lived tracers as variables in the regression model. As expected, concentrations of long-lived tracers seem to show close relationships to the biases seen in

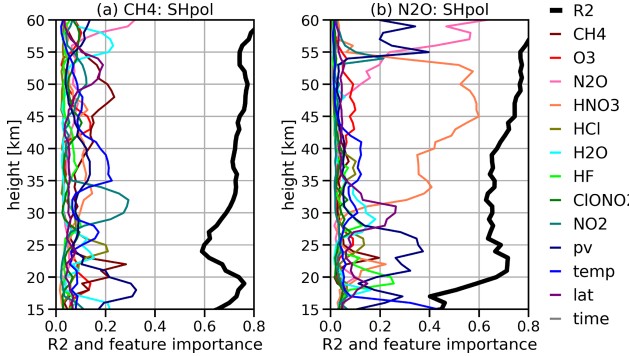

**Figure 1.** Vertical profiles of the variance ($R^2$) and feature importances estimated by XGBoost regression models for the TOMCAT–observation differences for **(a)** CH$_4$ (1991–2018) and **(b)** N$_2$O (2004–2018, ACE only) for the SHpol (50–90° S) latitude bin. See Eq. (1) and subsequent information about the features (13 in total) for variables used in the XGBoost regression model.

CH$_4$ and N$_2$O profiles. However, Figs. S1 to S4 show that $R^2$ values for other latitude bins are somewhat smaller (near 0.5), indicating that regions with less dynamical variability (e.g. mid-latitudes) might need some additional features that are not included in this setup.

Another important aspect is that $R^2$ values for CH$_4$ remain almost flat between 25 to 50 km, but for N$_2$O, $R^2$ values are close to 0.6 in the lower stratosphere and lower mesosphere, with minima near 30 km. The time (date) term is included in the XGBoost model to allow it to extrapolate corrections to data that lie outside the training period. However, in the current setup, the feature importance of the time term is only significant at a few levels for some latitude bands (Figs. 1 and S1 to S4). This suggests that the time term does not play a major role in the model's predictions for these latitude bands. To improve the model's performance, we also tried to increase the number of trees, using Huber/quantile loss functions, but none of the changes helped to improve the significance of the time term. In summary, in the current setup, the time (date) term is not very important.

In Fig. 1, dynamical variables such as potential vorticity are most important in explaining the biases in the lower stratosphere (near 18 km). This is likely due to the fact that the TOMCAT model in the setup employed here overestimates the fast isentropic transport in the lower stratosphere. However, it is important to note that the 50–90° S region covers a large part of the high-latitude stratosphere and includes the strong wintertime polar vortex as well as tracer variations near the edge of the vortex. As a result, it is not possible to attribute the biases to a single variable or process. For example, temperature variations explain a large part of the CH$_4$ biases around 35 to 40 km, but ClONO$_2$ is most important just below 35 km. On the other hand, HNO$_3$ is most important for explaining the N$_2$O biases in the mid-upper stratosphere. This suggests that, while there is a strong relationship

between temperature, potential vorticity and chlorine activation, the biases in CH$_4$ and N$_2$O at a single level are generally better explained by a single variable.

For CH$_4$, additional features showing significant importances are water vapour, CH$_4$ and N$_2$O. As CH$_4$ is the largest in situ source of stratospheric water vapour, their alternating importances in the lower mesosphere (above 55 km) indicate issues with HO$_x$-related CH$_4$ loss in the lower mesosphere. On the other hand, in the lower stratosphere, a strong wintertime dehydration inside the polar vortex leads to substantial drying. Hence, the somewhat larger importance for water vapour near 15 and 23 km suggests that XGBoost is able to identify and attribute possible biases in the TOMCAT setup to the downward transport of CH$_4$ as well as the parameterized dehydration scheme. Similarly, the peaks in N$_2$O importance near the stratopause ($\sim$ 48 km) and near 21 km indicate issues in the representation of the downward transport of the long-lived tracers from the mesosphere into the stratosphere in the polar vortex. Note that, in our simulations, the TOMCAT top model level is located near 60 km.

Next we compare vertical CH$_4$ profiles from TOMCAT, TCOM-CH4 and collocated HALOE/ACE for the SHpol latitude bin (Fig. 2). Overall, we have about 40 000 profiles, of which around 30 000 fall in the XGBoost training period and about 10 000 TS2 profiles fall in the 2019–2021 evaluation period. Overall, TCOM-CH4 profiles show excellent agreement with satellite profiles, and median lines seem to follow each other very closely. In contrast, the TOMCAT profiles show good agreement with observational data between 20–30 km but exhibit positive biases at upper and lower levels. This distinct feature indicates a clear separation in the importance of dynamical and chemical processes controlling CH$_4$ concentrations. As mentioned earlier, positive biases in TOMCAT CH$_4$ in the lower stratosphere could be due to faster CH$_4$ transport from the tropics to high latitudes. Positive biases in the upper stratosphere/lower mesosphere are most probably due to slower CH$_4$ loss via HO$_x$ and ClO$_x$ chemistry. Another important characteristic of Fig. 2 is that the variability in observational profiles (the shaded regions show 10th to 90th percentile ranges) is much larger than that in the TOMCAT (or TCOM) profiles. A possible explanation for differences in variability would be that the model output is sampled at the longitude/latitude recorded at 30 km tangent height, but in reality collocations at different altitudes are a few degrees apart. Additionally, the onion-peeling algorithm used for some solar occultation measurements (such as SAGE, HALOE) assumes that observations at different tangent heights are independent; hence, retrieved profiles show larger fluctuations.

Vertical profiles of the absolute (in ppm) and percentage (%) CH$_4$ differences between the three data sets are also shown in Fig. 2 for both the training (1992–2018) and evaluation (2019–2021) time periods. As expected, the median TCOM-CH4 profiles show very little difference with respect to collocated median satellite profiles, whereas the TOM-

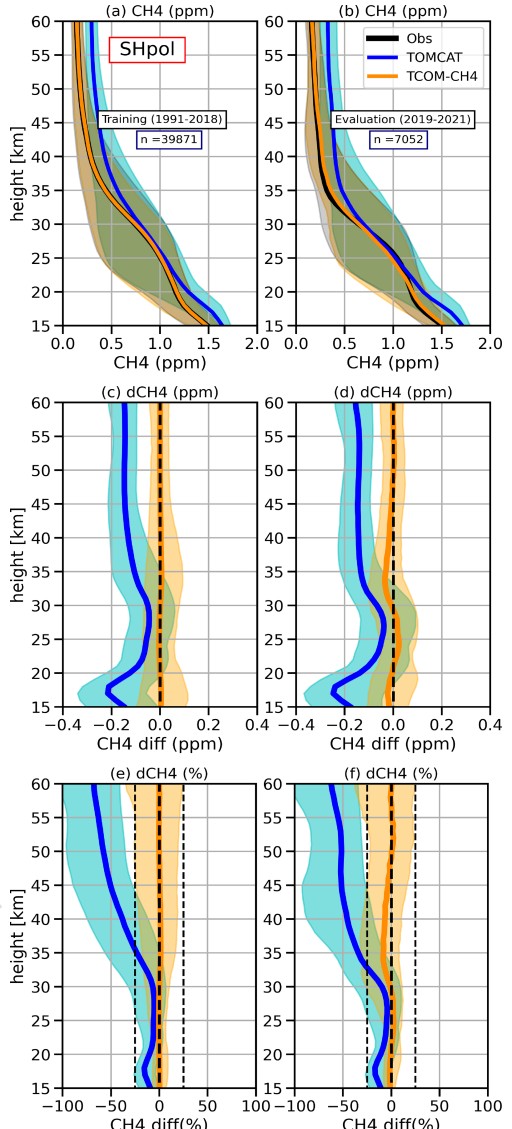

**Figure 2. (a, b)** Comparison between TOMCAT (blue), TCOM-CH4 (orange) and satellite-measurement-based (black) CH$_4$ profiles for the SHpol (50–90° S) latitude band. Solid lines indicate median profiles, while shaded regions show the 10th to 90th percentile range. Comparisons are shown for training (1992–2018) and evaluation (2019–2021) periods in panels **(a)** and **(b)**, respectively. Panels **(c)**–**(f)** show differences between TOMCAT and TCOM-CH4 w.r.t. satellite data sets in absolute units (ppm) and percent. Left **(c, e)** and right **(d, f)** panels show differences for the training (1992–2018) and evaluation (2019–2021) periods.

CAT profile differences range from $-0.22$ ppm (16 km) to $-0.05$ (near to 28 km). In terms of relative differences, again TCOM–observation differences are close to 0 %, whereas for the evaluation period, differences are up to 10 % in the lower and middle stratosphere. A possible explanation for somewhat larger differences for the 2019–2021 time period is that there has been rapid increase in atmospheric CH$_4$ over the

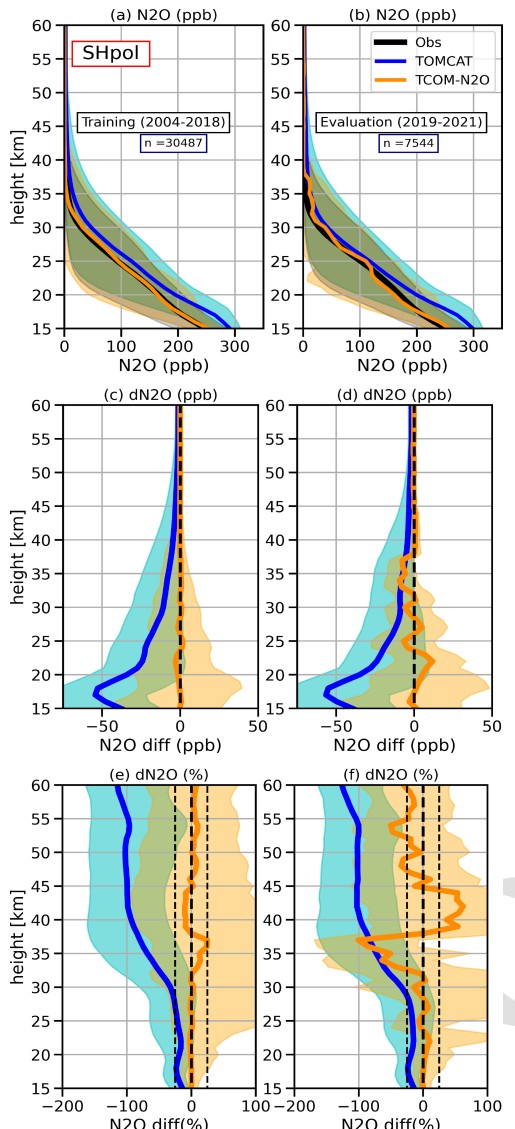

**Figure 3.** Same as Fig. 2 but for N$_2$O. The training period includes data for 2004–2018.

last few years (e.g. Nisbet et al., 2019). As the period of rapidly increasing CH$_4$ is outside the XGBoost training values, the estimated correction terms seem to be too small, but there are still significant improvements compared to TOM-CAT profiles. Median profile comparisons for training and evaluation periods and subsequent differences (in ppm and %) for other latitude bins are shown in Figs. S5 to S8. Again, the TCOM–observation comparison is also consistent for other latitude bins, with an exception that mid-stratospheric biases for the evaluation period are somewhat larger (up to 10 %) for the SHmid and the tropics (Figs. S5 and S6).

Similarly, for N$_2$O, Fig. 3 compares median profiles from ACE-FTS N$_2$O, TCOM-N2O and TOMCAT and their differences (absolute and percentage) for SH polar latitudes. Again, the TCOM-N2O and ACE-FTS profiles show excel-

lent agreement for all stratospheric altitudes. Interestingly, the TOMCAT N$_2$O profiles are high-biased only in the lower stratosphere (up to 25 km); they have negligible biases in the mid-upper stratosphere. So, in the lower stratosphere, the TOMCAT profiles show positive biases of up to ±50 ppb, while TCOM-N2O biases are close to zero for the training period (2004–2010) but show a slight increase (up to ±10 ppb) for the evaluation period (2019–2021). Some of these biases could be linked to the use of measurements with positive values only, and it is possible that there is a missing variable that accounts for strong seasonal variations at higher latitudes in the current setup for XGBoost. Although TCOM-N2O biases are much smaller throughout the stratosphere, in percentage terms they can reach up to 100 % near to 40 km as changes in the small values can translate into much larger changes in relative differences. However, even with those large relative differences, significantly reduced biases in TCOM-N2O profiles are visible for all the levels.

Median profile comparisons and differences between ACE-FTS, TCOM-N2O and TOMCAT profiles (in ppb and %) for other latitude bins are shown in Figs. S9 to S12. Similar to SHpol, the absolute median differences between observed and TCOM values for other latitude bins are less than 10 ppb. However, the relative differences in the upper stratosphere are much larger (up to 100 %, especially in the SHmid and tropics). This is likely due to the fact that TCOM only uses positive values, which removes observed profiles with low concentration values during the winter months.

Improvements in CH$_4$ and N$_2$O profiles are clearly visible in the time series comparisons shown in Figs. 4 and 5, which compare CH$_4$ and N$_2$O evaluations at 20, 30, 40 and 50 km for the SHpol latitude bin. For clarity, the figure shows every 10th profile (10 % of the data points). Similar comparisons for SHmid, the tropics, NHmid and NHpol are shown in Figs. S13 to S20. TCOM-CH4 data points show excellent agreement with the HALOE and ACE data points (Fig. 4). Uneven data density before and after 2004 reflect differences in viewing techniques between these two satellite instruments. Basically, HALOE was designed to provide near-global coverage, whereas ACE-FTS was designed to provide denser coverage at high latitudes. Even with these uneven sampling frequencies, we do not observe any abrupt changes in TCOM-CH4 data points.

Similarly for N$_2$O, Fig. 5 also shows excellent agreement between TCOM-N2O and ACE-FTS data points. Again, the largest corrections are observed in the lower stratosphere (15 to 25 km), where TOMCAT profiles are about 30 ppb high-biased, which can be considered a systematic bias due to the TOMCAT setup. Similar to CH$_4$, a seasonal minimum occurs just after the break-up of the Antarctic polar vortex (October) as the descending branch of stratospheric circulation transports N$_2$O-depleted air to lower altitudes and latitudes (horizontal mixing). As N$_2$O mixing ratios decrease rapidly with increasing altitude, a large number of ACE-FTS data points show negligible N$_2$O values, which is reflected in

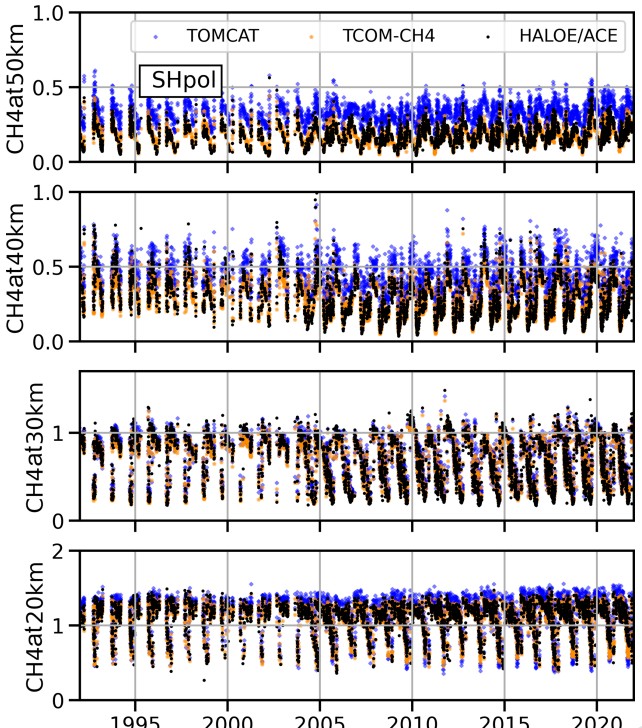

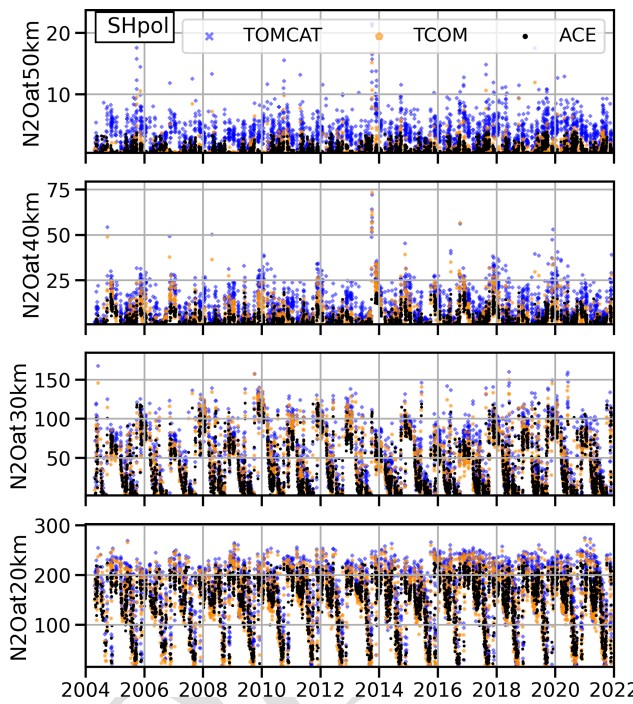

**Figure 4.** Time evolution (1992–2021) of CH$_4$ (in ppm) from TOMCAT (blue crosses), TCOM-CH4 (orange diamonds) and satellite data (black dots) for SHpol (50–90° S) at 20, 30, 40 and 50 km. Note that, for clarity, only 10 % of the data points (every 10th point) are shown. Due to the sharp gradient in the vertical distribution, the $y$ axis range varies between the panels.

**Figure 5.** Same as Fig. 4 but for N$_2$O (in ppb). The comparison is shown for the 2004–2021 time period.

the TCOM-N2O data points. However, it is also important to note that both CH$_4$ and N$_2$O mixing ratios decrease rapidly with increasing altitude (especially during SH autumn/winter). As the forward model used in the ACE-FTS retrieval algorithm needs spectra at fixed height levels, a seasonal variation in the vertical structure of the atmosphere alters the spacing between tangent heights. Therefore, N$_2$O (as well as CH$_4$) profile variability increases when tangent heights get very close together. Additionally, as mixing ratio values get close to zero, retrieved profiles become noisy, as some values can be negative. Here, we use only positive data points for XGBoost training, so the correction terms used here might be positively biased, influencing the seasonal cycle effects on CH$_4$ and N$_2$O concentrations.

Another important feature seen in Figs. 4 and 5 is that the seasonal cycles in TCOM-CH4 and TCOM-N2O data points seem to be more synchronized with observational data sets than TOMCAT, especially at 20 km. As shown above, TOMCAT profiles show positive biases throughout the stratosphere, and the largest corrections seem to be in the summertime maximum values that must arise from transport from mid-high latitudes. Interestingly, near to 30 km, points from all three data sets seem to be closer to each other for both

CH$_4$ and N$_2$O. Finally, an interesting aspect of both Figs. 4 and 5 is that in the upper stratosphere, both species show wintertime minima near to 40 to 50 km that are close to zero throughout the data record. Even with long-term trends in tropospheric concentrations, a casual inspection does not show any significant trends in either species. We aim to explore this aspect of CH$_4$ and N$_2$O trends in future studies.

Next, we compare TCOM-CH4 profiles with the latest SPARC CH$_4$ data set (Hegglin et al., 2021). Figure 6 show daily mean zonal mean CH$_4$ time series from TCOM-CH4 and monthly mean values from three SPARC (S-HALOE-CH$_4$, S-MIPAS-CH$_4$ and S-ACE-CH$_4$) CH$_4$ data records. Unsurprisingly, with some exceptions (near to 32.5° S and N), TCOM-CH4 shows the best agreement with S-ACE-CH$_4$ data at all pressure levels and latitude bins. At 50 hPa, TCOM-CH4 values show somewhat positive biases with respect to S-HALOE-CH$_4$ near subtropical latitudes but better agreement in the middle (5 hPa) and upper (0.5 hPa) stratosphere. On the other hand, S-MIPAS-CH$_4$ data points show significant positive biases with respect to all other data records, with qualitative agreement in the upper stratosphere. Additionally, as expected, positive growth rates observed in the tropospheric CH$_4$ concentrations are also distinguishable in both observations (ACE + HALOE) and TCOM-CH4 data, especially near tropical and subtropical latitudes at 50 hPa.

Figure 6 also compares the CH$_4$ evolution at 67.5° S and 67.5° N. As expected, wintertime CH$_4$ concentrations in the

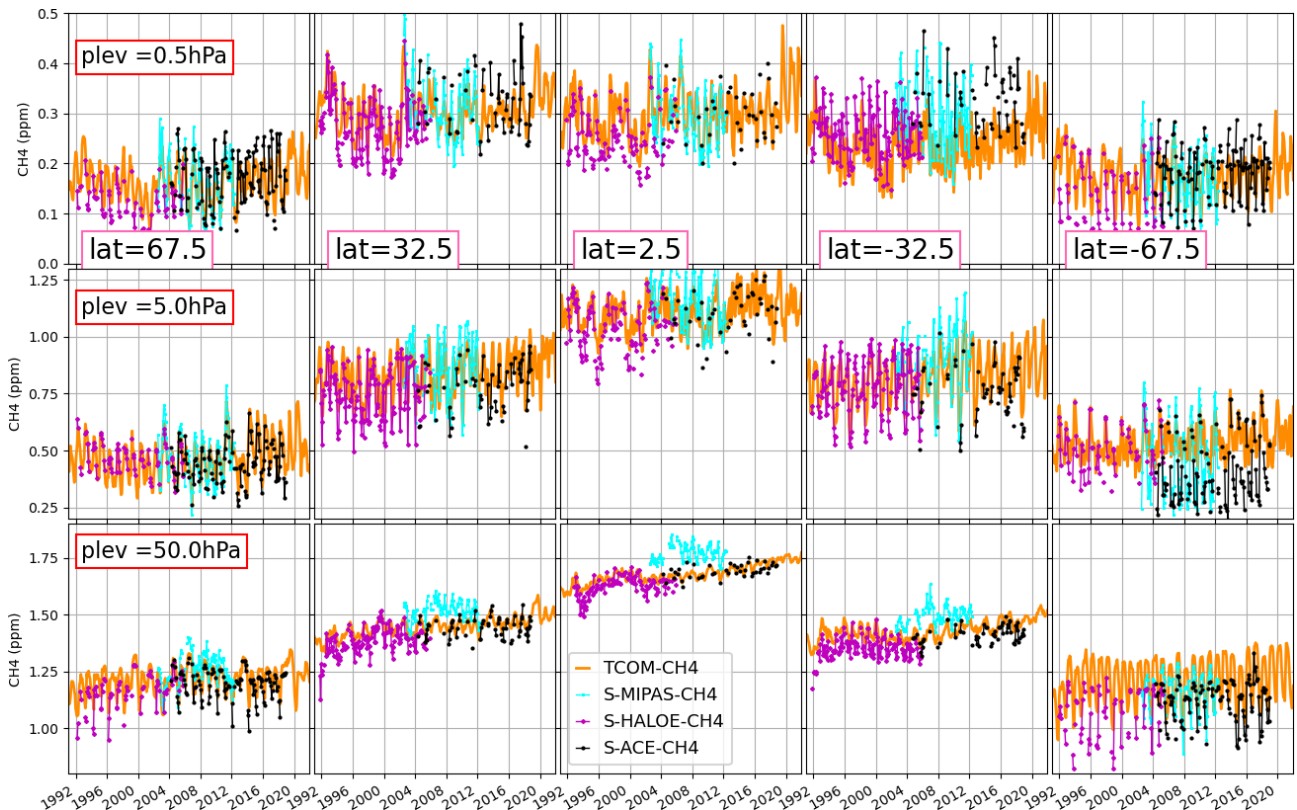

**Figure 6.** Comparison between TCOM-CH4 (orange line) and three (ACE (black line), HALOE (magenta line) and MIPAS (aqua line)) satellite instrument-based SPARC-CH₄ data sets (for details see Hegglin et al., 2021). Time series of monthly mean values from SPARC-CH₄ and TCOM-CH4 data sets are shown for 0.5 hPa (top), 5 hPa (middle) and 50 hPa (bottom) for five latitude bins: 67.5° and 32.5° in both hemispheres as well as 2.5° N (middle).

SH high latitudes are somewhat larger compared to those in the NH high latitudes (e.g. Remsberg, 2015). This is because a stronger Brewer–Dobson (BD) circulation in the NH causes faster downward propagation of the CH₄-poor air from the upper stratosphere to the lower-middle stratosphere. Interestingly, all the SPARC CH₄ data records show less CH₄ in the SH at 50 hPa than TCOM. At 5 hPa, TCOM-CH4 data show better agreement with S-HALOE-CH₄ data in both hemispheres and with S-ACE-CH₄ data only in the NH. The exact causes of the unusually low CH₄ values in S-MIPAS-CH₄ and S-ACE-CH₄ data files are unclear. A possible explanation might be that negative data points seen in ACE data (due to enhanced wintertime downwards transport of CH₄-poor air) are excluded in the XGBoost training step. It also suggests that wintertime downward descent at higher latitudes is somewhat weaker in TCOM data. Again, S-MIPAS-CH₄ data points indicate a much larger magnitude of the seasonal cycle compared to other data sets. In the upper stratosphere (0.5 hPa), there seems to better agreement among all the data in both hemispheres. Overall, we find that, compared to the TCOM-CH4 data set, SPARC CH₄ data records have some inconsistent characteristics, and the largest disagreement is found to occur at NH high latitudes.

Figure 7 compares the evolution of TCOM-N2O and SPARC data sets based on MIPAS, Aura-MLS, SMR and ACE measurements for five latitude grids (67.5° S, 32.5° S, 2.5° N, 32.5° N and 67.5° N) and three pressure levels (50, 5 and 0.5 hPa). The lack of satellite measurements during the 1990s makes it difficult to compare the long-term N₂O evolution, but significant differences between various satellite data records also complicate the more straightforward evaluation. Overall, TCOM-N2O shows the best agreement with SPARC ACE-FTS (S-ACE-N₂O) data and the poorest agreement with SPARC MIPAS (S-MIPAS-N₂O) data. Interestingly, SPARC SMR (S-SMR-N₂O) shows N₂O variations that are very similar to the S-MIPAS-N₂O data set, whereas SPARC-Aura-MLS (S-AMLS-N₂O) agrees better with S-ACE-N₂O, with some exceptions in the later few years that are related to a drift in the MLS N₂O measurement (190 GHz) channel (Livesey et al., 2021), especially in the lower stratosphere. Hence, for the earlier period, TCOM-N2O also shows good agreement with S-AMLS-N₂O data until 2014, and then slight drifts are distinguishable at low-mid latitudes. On the other hand, the close agreement between S-SMR-N₂O and S-MIPAS-N₂O means that both data

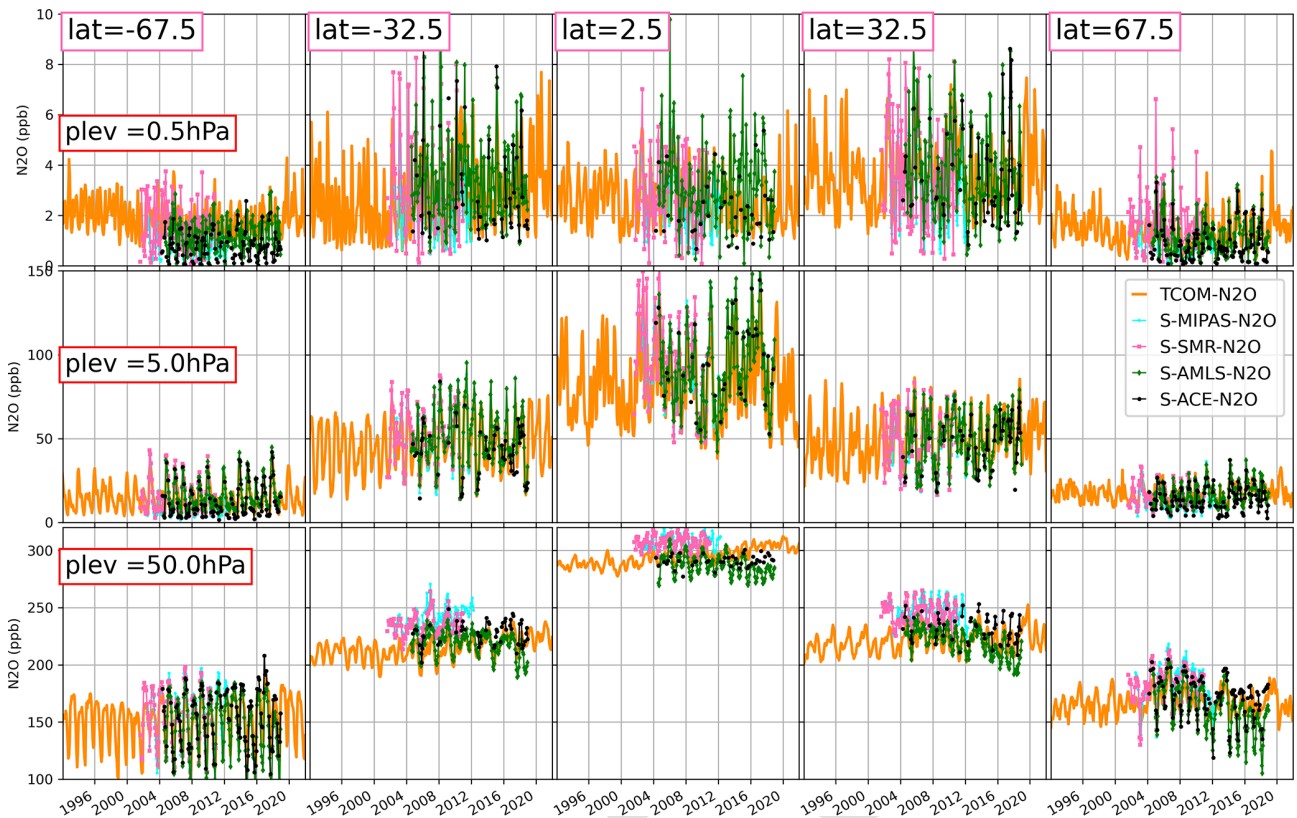

**Figure 7.** Same as Fig. 6 but for $N_2O$. SPARC data from the four satellite instruments ACE (v3.6), Aura-MLS (v4), MIPAS (v422) and SMR (v2.1) are shown with black, green, aqua and pink lines, respectively.

sets exhibit high biases in the lower stratosphere, and again agreement is weakest at low-mid latitudes.

Another important aspect of Fig. 7 is that at high latitudes, as well as for mid-upper stratospheric altitudes, all the SPARC data sets agree quite well with each other, and there are no long-term drift and systematic biases between them. The good agreement of TCOM-N2O with all the SPARC $N_2O$ data sets at 5 and 0.5 hPa provides additional evidence of the usefulness of the TCOM-N2O data set. Additionally, similar to TCOM-CH4, casual inspection of TCOM-N2O does not show any long-term trends at mid-upper stratospheric pressure levels; we aim to investigate this further in future studies.

Next, we analyse differences between TCOM-CH4 and TOMCAT $CH_4$ profiles through the time evolution of corrections estimated by the XGBoost regression model. First, we look at the differences in zonal mean $CH_4$ at different levels. Figure 8 shows TCOM-CH4 minus TOMCAT $CH_4$ differences (in %) at four vertical levels (15 to 45 km with 10 km spacing). An important aspect regarding 15 km and 25 km differences is that although the median $CH_4$ differences shown in Fig. 2 indicate that TOMCAT profiles show positive biases (of up to 10 %), the latitude slice indicates significant variations between the two. Differences are even positive close to polar latitudes, indicating stronger down-

ward transport of $CH_4$-poor air and/or weaker mixing near the Antarctic polar vortex region in the TOMCAT simulation. Similar characteristics are observed at NH high latitudes. These biases are even more distinct at 25 km, especially in the SH high latitudes, though this region can be considered to be a boundary region where dynamical processes control $CH_4$ concentrations at lower altitudes and chemical processes dominate at higher altitudes. Inter-hemispheric asymmetry in the $CH_4$ bias correction also indicates significant differences in the representation of the BD circulation in ERA5 data (e.g. Li et al., 2022).

Additionally, some uneven differences for 1991–1993 at 15 and 25 km in Fig. 8 could be due to a combination of various chemical and dynamical processes. For example, volcanically enhanced stratospheric aerosol following the Mt Pinatubo eruption in June 1991 might have altered stratospheric transport pathways, as larger aerosols absorb outgoing long-wave radiation (Free and Lanzante, 2009; Dhomse et al., 2020). Such heating can also enhance tropical upwelling as well as horizontal mixing on isentropic surfaces (e.g. Poberaj et al., 2011; Dhomse et al., 2015; Bittner et al., 2016). Volcanically enhanced stratospheric aerosol can also alter OH radical concentrations either by modulating the amount of incoming solar radiation or by altering chemical pathways (e.g. Bândă et al., 2013, 2016). It is also

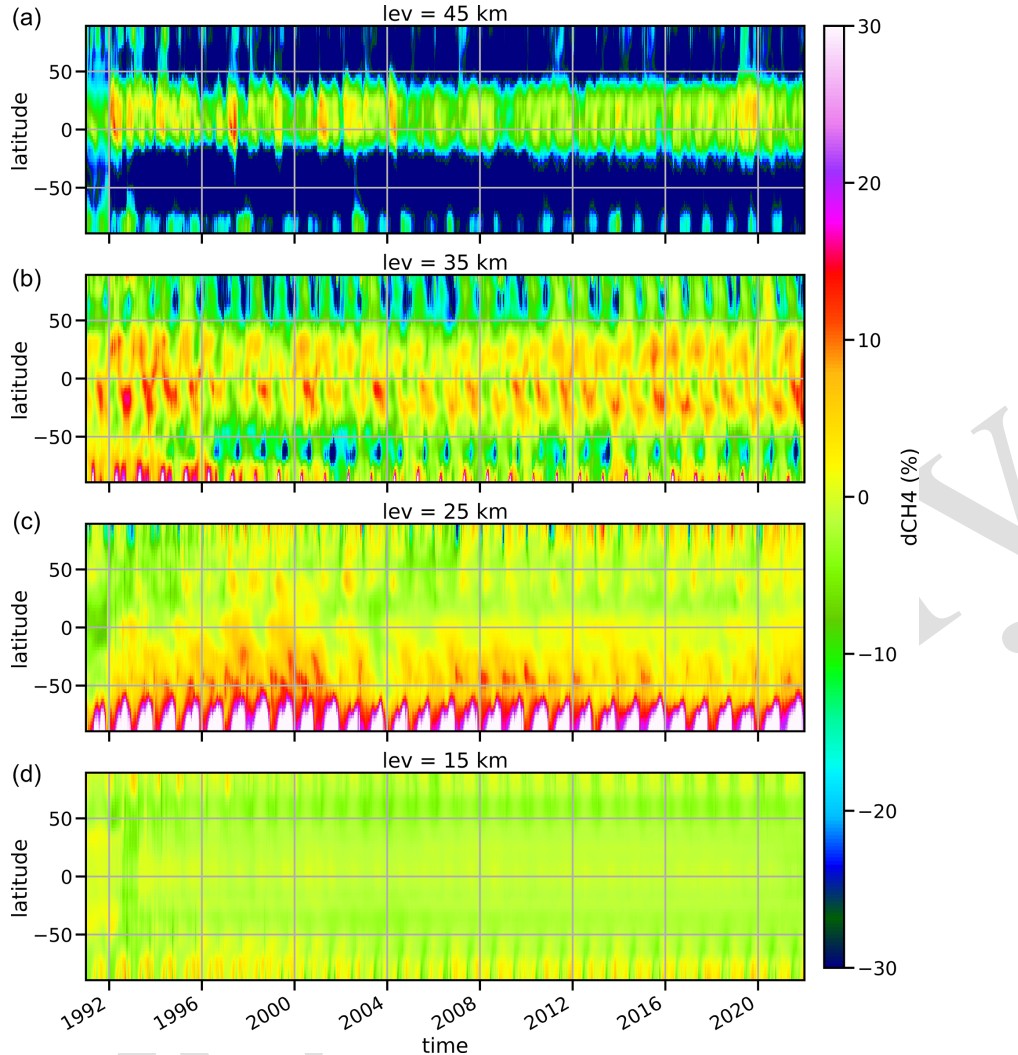

**Figure 8.** Latitude–time cross section of the differences between TCOM-CH4 and TOMCAT CTM CH₄ profiles (in %) at **(a)** 45 km, **(b)** 35 km, **(c)** 25 km and **(d)** 15 km. Percent differences are calculated as $200 \times (\text{TCOM} + \text{TOMCAT}) / (\text{TCOM} - \text{TOMCAT})$.

well known that increases in stratospheric aerosol concentration also affected HALOE retrievals (e.g. Remsberg, 2008). ERA5 data assimilation does not include these atmospheric effects of volcanically enhanced stratospheric aerosol (e.g. Hersbach et al., 2020); hence, we are not sure about the unusual CH₄ differences in the lower stratosphere.

On the other hand, differences at 35 km in Fig. 8 seem to be dominated by the quasi-biennial oscillation (QBO). QBO-induced meridional circulation patterns (e.g., Baldwin et al., 2001), which are underestimated in TOMCAT. Even though ACE provides limited observational data points in the tropics, XGBoost is able to identify this discrepancy. On the temporal scale, differences are largest until 1996, reaching polar latitudes, after which they gradually decrease in the NH subtropics and remain larger in the SH sub-tropics. A similar type of uneven evolution for later periods can also be seen, suggesting issues with the ERA5 data regarding the representation of QBO-induced circulation, especially for years with unusual QBO evolution such as 2016 and 2020 (e.g. Newman et al., 2016; Osprey et al., 2016; Diallo et al., 2022).

Another notable feature in Fig. 8 is that at 45 km, CH₄ differences are clearly distinguishable in some years. Both HALOE and ACE have much smaller retrieval errors at higher altitudes, and, assuming there were no abrupt changes in these two satellite instruments (or retrieval algorithms), the unusual differences seen at 45 km can be attributed to inhomogeneities in or issues with the ERA5 data. These distinctive periods include the first halves of years 1993, 1997, 2001 and 2004 and the latter half of 2019.

## 6 Data availability

HALOE V19 data are available from https://acdisc.gesdisc.eosdis.nasa.gov/data//UARS_HALOE_Level2/

(Russell and James, 1999). ACE-FTS v4.2 data are obtained via http://www.ace.uwaterloo.ca/data.php (last access: 13 January 2022) (DOI: https://doi.org/10.20383/101.0291, (Bernath et al., 2020)). SPARC climatological data can be obtained via https://doi.org/10.5281/zenodo.4265393 (Hegglin et al., 2020). TCOM-CH4 and TCO-N2O data are publicly available at https://doi.org/10.5281/zenodo.7293740 (Dhomse, 2022a) and https://doi.org/10.5281/zenodo.7386001 (Dhomse, 2022b), respectively.

## 7   Summary and conclusions

Even though CH$_4$ and N$_2$O are very important greenhouse gases, as well as the sources for key stratospheric species, there are limited stratospheric profile data sets that extend for more than a decade. Due to their long lifetimes, CH$_4$ and N$_2$O are also very useful dynamical tracers that can used to evaluate stratospheric transport processes. Also, accurate stratospheric CH$_4$ profiles are a valuable constraint for the retrieval of tropospheric methane using satellite instruments. However, until now, no attempt has been made to construct long-term CH$_4$ and N$_2$O profile data sets. Furthermore, although chemical models are able to simulate long-term profile data sets of these species, they are highly dependent on the representation of individual chemical and dynamical processes in a particular model.

Here we have used CH$_4$ and N$_2$O profiles from the TOMCAT CTM, two solar occultation instrument measurements, and the eXtreme Gradient Boost (XGBoost) regression model to construct daily, gap-free stratospheric profile data sets (TCOM-CH4 and TCOM-N2O) for the 1991–2021 time period. The XGBoost regression model is trained for the CH$_4$ or N$_2$O difference between TOMCAT and satellite measurements (HALOE and ACE). These differences are used to estimate corrections that are added to the TOMCAT profiles to derive TCOM-CH4 and TCOM-N2O profiles. The regression algorithm uses 13 features (or variables) based on TOMCAT tracers as well as four additional features such as temperature, potential vorticity, latitude and date of the measurement. Because atmospheric concentrations of CH$_4$ and N$_2$O vary due to distinct dynamical and chemical processes in different regions, our approach involves dividing global measurements into five latitude-based categories. These categories include two for the polar regions, two for the mid-latitudes, and one for the tropics. We then proceed to derive regression parameters for each 1 km vertical grid spanning from 15 to 60 km within each of these latitude bins.

For both gases considered, XGBoost shows good performance ($R^2 > 0.5$ to 0.8) throughout the stratosphere except for the lower stratosphere, which can be attributed to the limited training measurements. Measurements from the final 3 years (2019–2021) are used to evaluate TCOM-CH4 and TCOM-N2O profiles. Overall, TCOM-CH4 shows ex-cellent agreement with the evaluation profiles, and median differences are less than 10 %. Additionally, comparison with SPARC-CH$_4$ data suggests that SPARC-MIPAS profiles show some unrealistic behaviour, and SPARC-ACE and SPARC-HALOE do not show expected inter-hemispheric differences in lower stratospheric CH$_4$ differences (there is less CH$_4$ in the NH).

For TCOM-N2O, better agreement is again seen with respect to the S-ACE-N$_2$O data set and weaker agreement is observed with MIPAS data. TCOM-N2O also confirms the abnormal drift in the Aura-MLS v4.2 N$_2$O data (as used in the SPARC data set), especially at lower latitudes and altitudes (e.g. Livesey et al., 2021). A casual inspection of TCOM-CH4 and TCOM-N2O plots also suggests that, despite increasing surface values, there are near-negligible long-term trends in the upper stratosphere/lower mesosphere, which is consistent with Minganti et al. (2022). On the other hand, (Prather et al., 2023) analysed MLS V5 data and showed positive trends (of up to 15 %) in the tropical upper stratospheric N$_2$O, though they did not find NO production to be rising at similar rates. A possible explanation would be that stratospheric/mesospheric loss processes, probably caused by changes in the stratospheric circulation, are reducing the lifetimes of these GHGs. We aim to analyse these discrepancies in future studies. Finally, analysis of TCOM-CH4 and TOMCAT CH$_4$ profiles suggests that the representation of QBO-induced secondary circulation is not adequate in the CTM, and differences also reveal some temporal inhomogeneities in the ERA5 reanalysis data.

Presently, the TCOM-CH4 and TCOM-N2O V1.0 data set is ideally suited for the evaluation of CH$_4$ and N$_2$O chemistry and stratospheric transport processes in models. The TCOM-CH4 data set can also be used as realistic stratospheric profiles in a CH$_4$ profile/total column retrievals. Daily mean zonal mean TCOM-CH4 and TCOM-N2O profile data on pressure and altitude levels in mixing ratio units are publicly available via https://doi.org/10.5281/zenodo.7293740 (Dhomse, 2022a) and https://doi.org/10.5281/zenodo.7386001 (Dhomse, 2022b), respectively.

**Supplement.** The supplement related to this article is available online at: https://doi.org/10.5194/essd-15-1-2023-supplement.

**Author contributions.** SSD conceived and designed the study. MPC performed TOMCAT model simulations. SSD performed the analysis. SSD and MPC co-wrote the paper.

**Competing interests.** The contact author has declared that neither of the authors has any competing interests.

**Disclaimer.** Publisher's note: Copernicus Publications remains neutral with regard to jurisdictional claims made in the text, published maps, institutional affiliations, or any other geographical representation in this paper. While Copernicus Publications makes every effort to include appropriate place names, the final responsibility lies with the authors.

**Acknowledgements.** This work was supported by the NERC SISLAC (grant no. NE/R001782/1) and LSO3 (grant no. NE/V0011863/1) projects. We thank the HALOE and ACE-FTS science teams for the data sets. We thank the European Centre for Medium-Range Weather Forecasts for providing their analyses. TOMCAT simulations were performed on the UK national Archer and Leeds Arc4 high performance computing (HPC) system.

**Financial support.** This research has been supported by the Natural Environment Research Council (grant nos. NE/R001782/1, NE/V0011863/1, and NE/W004895/1) and ESA via OREGANO project (contract No. 4000137112/22/I-AG).

**Review statement.** This paper was edited by Guanyu Huang and reviewed by Chris Boone and two anonymous referees.

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

## Remarks from the typesetter