# Peer review of "Using machine-learning to construct TOMCAT model and occultation measurement-based stratospheric methane (TCOM-CH4) and nitrous oxide (TCOM-N2O) profile data sets"

_Earth System Science Data, 2023_

## Referee Comment (RC1)

**General Comments:**

The paper by Dhomse and Chipperfield describes two new stratospheric data sets (TCOM-CH4 and TCOM-N2O), which were generated by a combination of TOMCAT model data and occultation measurements using a machine learning approach.
These data sets are unique in a sense that different satellite instruments were used to generate the merged long-term stratospheric data set.
The data are in general useful – as also written in the paper – for the evaluation of models and as a-priori information for retrievals.
Both data sets are publicly available for download (has been checked).

The paper is well written and contains all relevant information for data users except for some points addressed below.

I have the following general comments/questions:

1. The provided data sets are zonal averages. However, the zonal averaging is not explained in the method. Is it done before or after training/application of the correction term?
   Please clarify in the paper.

2. It seems that in the machine learning approach the occultation measurements are considered as 'truth' for the training (and also later for testing/validation).
   How are uncertainties of the measurement data considered?
   How large are these, and how do they compare to the differences seen during testing/validation?
   Retrieval errors are mentioned in the paper, but only in a general way without any quantitative assessment.
   It is also unclear, if/how the vertical resolution of the measurements / averaging kernels etc. is considered in the method.
   This should be addressed in the paper.

3. How relevant is the time dependence in the regression model?
   Would it be possible to use this correction also for times not covered by the measurements (it seems so as the evaluation period is after the training period)?
   What do you think are the limitations?

**Specific Comments:**

1. p. 5, l. 121:
   Please explain SWOOSH.

2. p. 5, l. 130:
   Note: 1 km is probably the vertical sampling of the ACE profiles; the vertical resolution depends on the averaging kernels.

3. p. 5, l. 133:
   Limiting retrieval errors to <100% is a quite coarse filter. Since the correction method does not seem to consider measurement errors this may be an issue at least at higher altitudes.

4. p.5. l. 136ff:
   *A 10° latitudinal overlap between the bins is allowed...*

   From the definition of the bins, it seems that the overlap is 20 degrees?

   Furthermore, the explanation *...to include possible extreme variations in the training data set* is unclear – why is the variation in the non-extended bins not sufficient?

As I understand, the following sentence *Estimated differences for overlapping grids are averaged...* actually does not refer to this step of the method (calculation of differences) but to later merging of the data in step 6.

These sentences should be re-formulated to clarify the above.

5. p. 5, eq. 1:
Please specify what exactly is meant with 'time' in this equation. Is it the absolute measurement time (e.g. UTC time) or local time? In general, units should be specified for all quantities (e.g. does CH4 refer to number density or a volume mixing ratio?).

6. p. 6, l. 176:
*which might introduce homogeneities*
Do you mean inhomogeneities?

7. p. 7, l. 1ff:
Are the feature importances normalised or not?

8. p. 7, l. 196/197:
It seems that these sentences (esp. the value 18 km) refers to CH4 - please clarify in the text.

9. p. 7, l 199/200:
*Hence attributing a single variable or a single processes is not possible.*
The formulation is unclear (attributing to what?).
In fact a similar statement is at the end of this paragraph.
Please reformulate.

10. p. 8, l. 234/235:
Please add a bit of summary information about Figs. S5 to S8, e.g. if results are similar for the other latitude bins or not.

11. p. 8, l. 236ff and Fig. 3:
The vertical variation of the TCON-N2O profiles seems to be larger than in the original data (both observation and model), especially for the evaluation data set.
Please explain.

12. p. 8, l. 245:
Also, please add some general information about Figs. S9 to S12.

13. p. 8, l. 253:
What exactly is meant with *unusual data points in 2004*?
Do you refer to the larger values at 40 km?
Please clarify.

14. p. 9, l. 2:
*largest corrections are observed in the lower stratosphere*
This is the case for the absolute corrections, not the relative ones.

15. p. 9, l. 259ff:
Especially regarding N2O, how large is the error of the measurements at high altitudes / low concentrations? How reliable are the measurements at high altitudes?

16. p. 9, l. 261–263:
*As the ACE-FTS retrieval algorithm uses multiple micro-windows, there may be a seasonal shift in averaging kernels causing fluctuations in the retrieved profiles.*
Why do multiple micro-windows cause a seasonal shift?
Do you mean that different micro-windows are used during different seasons?
Please clarify.

17. p. 9, l. 263/264:
    *As we use only positive data points for XGBoost training...*
    Due to measurement uncertainties the occultation profiles may contain negative data points.
    If you only use positive data for training this might results in a bias.
    Please clarify.

18. p. 9, l. 274/275:
    Why do you show in Fig. 6 daily means for TCOM but monthly means for SPARC data?
    Wouldn't it be better to use for the comparisons in both cases the same averaging time
    interval?
    Please explain.

19. p. 9, l. 274ff:
    What is the difference between the ACE-FTS CH4 data set used in this study and the
    corresponding SPARC data set?
    Please explain.

20. p. 10, l. 310ff:
    Only a suggestion:
    Maybe the comparisons between TOMCAT and TCOM should be described before the
    validation with independent data sets as they define the expected accuracy of the TCOM
    data.

21. p. 11, l. 338 and Fig. 8:
    *for some years CH4 differences are clearly distinguishable.*
    Please be more specific here.
    Do you mean the occasionally high values in the tropics?
    Actually, a lot of differences at 45 km seem to be below the lower range of the colour scale
    (-15%).
    How representative are relative values at high latitudes where concentrations are low?
    What is the reference for the relative values (TCOM or TOMCAT)?

22. p. 12, l. 370:
    *A possible explanation would be strengthening of the stratospheric circulation...*
    What do you want to explain here? A trend or a non-existent / negligible trend (as mentioned
    in the previous sentence)?
    Please clarify.

**Technical Corrections:**

1. In general, please check the text for missing 'the' in the sentences.

2. p. 5, l. 125:
   occulatation → occultation

3. p. 5, eq. 1:
   I suggest that instead of the written quantity names (like 'temperature') variables should
   be used in this equation. Note that a sequence of italic letters in an equation could be
   (formally) misinterpreted as products of single variables.

4. p. 8, l. 229:
   to -0.05 → to -0.05 ppm.

5. Figs. 4 and 5:
   Please specify the CH4 and N2O unit in the figure or the caption.

6. Suggestion regarding the data sets:
   It would be good to have the unit of zonal mean CH4 and N2O not only in the global attributes of the data sets but also (or instead) in the attributes of the corresponding zonal mean variables.

---

## Author Comment (AC1)

Replies to Reviewer #1

General Comments: The paper by Dhomse and Chipperfield describes two new stratospheric data sets (TCOM-CH4 and TCOM-N2O), which were generated by a combination of TOMCAT model data and occultation measurements using a machine learning approach. These data sets are unique in a sense that different satellite instruments were used to generate the merged long-term stratospheric data set. The data are in general useful – as also written in the paper – for the evaluation of models and as a-priori information for retrievals. Both data sets are publicly available for download (has been checked). The paper is well written and contains all relevant information for data users except for some points addressed below.

*## We would like to thank the Reviewer #1 for his/her encouraging comments. Our replies are in blue italics. Briefly, we have done following changes (detailed response starts from page 2)*

- *Clarified the zonal averaging procedure.*

- *Discussed the uncertainties of the measurement data and how they are considered in the method.*

- *Explained that the time dependence in the regression model is not very significant for most latitude bands.*

- *Defined the term "SWOOSH".*

- *Corrected the errors in the text regarding the vertical resolution of the ACE profiles and the averaging kernels.*

- *Corrected the sentence "which might introduce homogeneities".*

- *Confirmed that the feature importances in the XGBoost model are not normalized.*

- *Reworded a paragraph with a sentence "Hence attributing a single variable or a single processes is not possible".*

- *Added a summary of the results from Figures S5-S8.*

- *Explained the reason for the larger vertical variation of the TCOM-N2O profiles in the evaluation data set.*

- *Added some general information about Figures S9-S12.*

- *Deleted the sentence about the unusual data points in 2004.*

- *Clarified that the largest corrections are observed in the lower stratosphere for the absolute corrections, not the relative ones.*

- *Discussed issues with high altitudes / low concentrations measurements.*

- *Clarified the reason why seasonal shift in the atmospheric structure, low concentrations, high beta angles contribute to nosier  retrieved profiles.*

- *Explained how only positive data points used for XGBoost training can affect the correction terms.*

I have the following general comments/questions:

1. The provided data sets are zonal averages. However, the zonal averaging is not explained in the method. Is it done before or after training/application of the correction term? Please clarify in the paper.

**We apologize for the confusion. In the revised manuscript, we clarify that we first calculate 3D (longitude/latitude/height) profiles twice a day (1:30 AM and 1:30 PM) before calculating the zonal mean. We then obtain the daily mean by averaging the 1:30 AM and 1:30 PM profiles.**

2. It seems that in the machine learning approach the occultation measurements are considered as 'truth' for the training (and also later for testing/validation). How are uncertainties of the measurement data considered? How large are these, and how do they compare to the differences seen during testing/validation? Retrieval errors are mentioned in the paper, but only in a general way without any quantitative assessment. It is also unclear, if/how the vertical resolution of the measurements / averaging kernels etc. is considered in the method. This should be addressed in the paper.

**We thank the reviewer for pointing out the lack of information. In the revised manuscript we added a paragraph to explain that we consider the measurements with positive values and retrieval error less than 100% to be an absolute truth and our attempt is to construct the data that would approximate HALOE/ACE if the instruments had denser measurements without any temporal gaps. We also clarify that we do not consider averaging-kernel-related information (ACE does not have averaging kernels) as it is impossible to get similar information for all the model grid points.**

3. How relevant is the time dependence in the regression model? Would it be possible to use this correction also for times not covered by the measurements (it seems so as the evaluation period is after the training period)? What do you think are the limitations?

**Indeed, the time (date) term is included in the XGBoost model to allow it to extrapolate corrections to data that lies outside the training period. However, in current setup, the feature importance of the time term is only significant at a few levels for some latitude bands. This suggests that the time term is not playing a major role in the model's predictions for these latitude bands. To improve model's performance, we also tried to increase number of trees, use Huber/quantile loss functions, but none of the changes helped to improve time term's significance. We have added discussion in a revised manuscript. In summary, in a current setup time (date) term is not very significant.**

Specific Comments:
1. p. 5, l. 121: Please explain SWOOSH.
**Done**

2. p. 5, l. 130: Note: 1 km is probably the vertical sampling of the ACE profiles; the vertical resolution depends on the averaging kernels.

*## Reviewer #2 (Dr Boone) correctly pointed out that ACE does not use averaging kernels. The forward model used in V4.2 retrieval uses 1 km vertical resolution, hence fitted spectra are interpolated at 1 km resolution.*

3. p. 5, l. 133: Limiting retrieval errors to <100% is a quite coarse filter. Since the correction method does not seem to consider measurement errors this may be an issue at least at higher altitudes.
*## Yes, at higher altitudes it can add some biases, but the median profiles seem to be close to the median profiles from observational data. As mentioned by Reviewer #2, it influences correction term estimates at high (low values throughout the year) and low altitudes (winter/spring time minima). We are aware that we cannot construct perfect data sets but our aim is to construct gap-free data set but similar to ACE/HALOE profiles.*

4. p.5. l. 136: A 10-degree latitudinal overlap between the bins is allowed… From the definition of the bins, it seems that the overlap is 20 degrees? Furthermore, the explanation …to include possible extreme variations in the training data set is unclear – why is the variation in the non-extended bins not sufficient?  As I understand, the following sentence Estimated differences for overlapping grids are averaged… actually does not refer to this step of the method (calculation of differences) but to later merging of the data in step 6. These sentences should be re-formulated to clarify the above.
*## We have expanded that discussion and clarified that we use 20-degree overlapping.*

5. p. 5, eq. 1: Please specify what exactly is meant with 'time' in this equation. Is it the absolute measurement time (e.g. UTC time) or local time? In general, units should be specified for all quantities (e.g. does CH4 refer to number density or a volume mixing ratio?).
*## Done. Time is measurement date, and all the tracers are in volume mixing ratio units.*

6. p. 6, l. 176: which might introduce homogeneities Do you mean inhomogeneities?
*## Corrected.*

7. p. 7: Are the feature importances normalised or not?
*## No, these are directly from XGBoost.*

8. p. 7, l. 196/197: It seems that these sentences (esp. the value 18 km) refers to CH4 - please clarify in the text.
*## Done.*

9. p. 7, l 199/200: Hence attributing a single variable or a single processes is not possible. The formulation is unclear (attributing to what?). In fact a similar statement is at the end of this paragraph. Please reformulate.
*## Done.*

10. p. 8, l. 234/235: Please add a bit of summary information about Figs. S5 to S8, e.g. if results are similar for the other latitude bins or not.
*## Done, again highlighting that biases are largest for SHmid and tropical latitude bands.*

11. p. 8, l. 236ff and Fig. 3: The vertical variation of the TCOM-N2O profiles seems to be larger than in the original data (both observation and model), especially for the evaluation data set. Please explain.

*## We have revised the manuscript to highlight issue with the use of only positive values for especially for the regions where concentrations are very low (especially upper stratosphere and lower mesosphere). We have also noted the lack of an explanatory variable that accounts for the strong winter/springtime seasonal minima at polar latitudes as downward transport brings N2O-poor air from mesosphere to the stratosphere.*

12. p. 8, l. 245: Also, please add some general information about Figs. S9 to S12.

*#Done. Again, highlighting the larger biases in the SHmid and tropics.*

13. p. 8, l. 253: What exactly is meant with unusual data points in 2004? Do you refer to the larger values at 40 km? Please clarify.

*## We have reviewed this issue once again and looks like those points are not unusual as we see similar features for other years. Therefore, we have decided to delete the sentence.*

14. p. 9, l. 2: largest corrections are observed in the lower stratosphere This is the case for the absolute corrections, not the relative ones.

*## Yes, we revised the sentence to clarify it and added that those biases can be considered as systematic bias due to TOMCAT set up.*

15. p. 9, l. 259ff: Especially regarding N2O, how large is the error of the measurements at high altitudes / low concentrations? How reliable are the measurements at high altitudes?

*## See replies to the earlier comments. Also, as correctly pointed by the reviewer and Reviewer #2, at lower concentrations and closer tangent heights at lower altitudes for high beta angle measurements means some ACE retrievals converge for negative values. However, the black data points shown in Figures 4 and 5 are the ones with positive retrieval values and retrieval errors less than 100%.*

16. p. 9, l. 261–263: As the ACE-FTS retrieval algorithm uses multiple micro-windows, there may be a seasonal shift in averaging kernels causing fluctuations in the retrieved profiles. Why do multiple micro-windows cause a seasonal shift? Do you mean that different micro-windows are used during different seasons? Please clarify.

*## As explained by Reviewer #2, ACE does not use averaging kernels, but the forward model uses 1 km tangent height spacing and at lower concentrations (and high beta angles) they might be shifted close to each other. So, we have reworded those sentences as:*
*"As the ACE-FTS retrieval algorithm uses multiple micro-windows, a seasonal variation in vertical structure of the atmosphere means interpolated radiances would have very little variations when N2O/CH4 concentrations are low.  Also, when concentrations of a gas low measured spectra would show very little change between two tangent heights, leading noisy profiles. Therefore, N2O (as well as CH4) profiles show large variability at variability increases when tangent heights get very close together. Additionally, as mixing ratio values get close to zero, retrieved values can be negative. Here, we use only positive data points for XGBoost training, so that correction terms used here might be positively biased, influencing seasonal cycle effects in CH4 and  N2O concentrations."*

17. p. 9, l. 263/264: As we use only positive data points for XGBoost training... Due to measurement uncertainties the occultation profiles may contain negative data points. If you only use positive data for training this might results in a bias. Please clarify.
*## Yes, Reviewer #2 also pointed out this issue. In the revised manuscript we added a discussion and mentioned that this might cause some positive biases in TCOM profiles.*

18. p. 9, l. 274/275: Why do you show in Fig. 6 daily means for TCOM but monthly means for SPARC data? Wouldn't it be better to use for the comparisons in both cases the same averaging time interval? Please explain.
*## We agree with the reviewer. Our aim was to show that TCOM data is available on a daily frequency, but for a direct comparison, we agree that we should have shown monthly means. The updated Figure 6 includes monthly means.*

19. p. 9, l. 274ff: What is the difference between the ACE-FTS CH4 data set used in this study and the corresponding SPARC data set? Please explain.
*## The main difference is that the SPARC data set uses ACE v3.6 data whereas here we use ACE v4.2 data. We aim to release TCOM 1.1 data that will use ACE v5.2 data and use both positive and negative values to avoid possible causes for the positive biases seen at higher latitudes and altitudes. We also note that SPARC data uses somewhat earlier versions of Aura-MLS (v4) and MIPAS (v422).*

20. p. 10, l. 310ff: Only a suggestion: Maybe the comparisons between TOMCAT and TCOM should be described before the validation with independent data sets as they define the expected accuracy of the TCOM data.
*## We have briefly expanded the discussion about the differences between TCOM and TOMCAT in the text. We have also expanded the discussion of the observation-TCOM differences, especially for evaluation period but translating this to the expected accuracy cannot be justified statistically, so we have refrained from doing so.*

21. p. 11, l. 338 and Fig. 8: for some years CH4 differences are clearly distinguishable. Please be more specific here. Do you mean the occasionally high values in the tropics? Actually, a lot of differences at 45 km seem to be below the lower range of the colour scale (-15%). How representative are relative values at high latitudes where concentrations are low? What is the reference for the relative values (TCOM or TOMCAT)?
*## We agree at higher altitudes absolute values are much smaller, hence percentage differences may not provide enough information. We have added a caution note in the revised manuscript. We also reiterate that we have a limited number of ACE profiles in the tropics which is reflected in smaller R2 values. We also added a sentence in the caption: "Differences are calculated as 200\*(TCOM-TOMCAT)/(TCOM+TOMCAT)" and altered the contour range to -30% to +30% so that larger differences are clearly distinguishable.*

22. p. 12, l. 370: A possible explanation would be strengthening of the stratospheric circulation... What do you want to explain here? A trend or a non-existent / negligible trend (as mentioned in the previous sentence)? Please clarify.
*## We have reworded the sentence to mention that positive trends in the tropospheric emission should increase stratospheric concentrations, but if it is compensated by the stratospheric/mesospheric losses then it would lead to much smaller trends in the stratospheric N2O.*

Technical Corrections:

1. In general, please check the text for missing 'the' in the sentences.
*## Done.*

2. p. 5, l. 125: occulatation → occultation
*## Done*

3. p. 5, eq. 1: I suggest that instead of the written quantity names (like 'temperature') variables should be used in this equation. Note that a sequence of italic letters in an equation could be (formally) misinterpreted as products of single variables.
*## Done*

4. p. 8, l. 229: to -0.05 → to -0.05 ppm.
*## Done*

5. Figs. 4 and 5: Please specify the CH4 and N2O unit in the figure or the caption. 3
*## Done*

6. Suggestion regarding the data sets: It would be good to have the unit of zonal mean CH4 and N2O not only in the global attributes of the data sets but also (or instead) in the attributes of the corresponding zonal mean variables.
*## We are sorry for the mistake. To avoid duplication of the data files, we aim to release v1.1 data that will extend until December 2022 with some minor updates such as using ACE v5.2 data and those files will have correct global and variable attributes.*

---

## Author Comment (AC2)

Review of "Using machine-learning to construct TOMCAT model and occultation measurement-based stratospheric methane (TCOM-CH4) and nitrous oxide (TCOM-N2O) profile data sets" by Sandip Dhomse and Martyn Chipperfield.

Overall, this is a well written and well-considered construction of long-term data sets for two important atmospheric molecules. I just have a few concerns that should be relatively easily addressed.

*## We would like to thank Dr Boone (Reviewer #2) for his insightful comments on our manuscript. His feedback has been invaluable in helping us to improve the clarity, accuracy, and overall quality of the paper. Our replies are in blue italic text.*

> Line 225: "Additionally, the onion peeling algorithm used for solar occultation measurements assumes observations at different tangent height are independent, hence retrieved profiles show larger fluctuations."
HALOE used onion peeling in its retrieval, but ACE-FTS does not. That is not the reason the variability is so high here. In this latitude region, you will see effects from atmospheric descent (the entire profile sinks to lower altitude in the stratosphere) inside the polar vortex during the winter. This would account for the pronounced 'bulge' in the variability around 20-25 km relative to the "trop" set, for example, which is the only region here that does not include a contribution from atmospheric descent. Additionally, in the lower stratosphere, you will see variability in CH4 from H2O-related chemistry, but most of the large variability seen in the data around 25-30 km presumably results from the inclusion of profiles experiencing different degrees of atmospheric descent inside the polar vortex over the course of the winter. It is a real, physical variability, not a retrieval artefact.

*## Yes, we agree. We have removed the sentence about the retrieval errors and averaging kernels and focussed more on wintertime downward descent leading to seasonal changes in tangent height might be causing dynamically forced strong annual cycle. We also highlight differences between two retrieval algorithms.*

> Line 231: "…somewhat larger differences for 2019-2021 time period is that there has been rapid increase in atmospheric CH4 over the last few years (e.g. Nisbet et al., 2019)."
Note that all the evaluation period (2019-2021) CH4 comparisons exhibit a bump around 35 km, where ACE-FTS results are slightly higher than the TCOM-CH4 results. This appears to coincide with the observation of larger trends at higher altitudes in ACE-FTS results that seem to result from a less efficient conversion of CH4 to H2O in the middle stratosphere in recent years (which leads to higher levels of CH4 in later years): doi:10.1016/j.jqsrt.2020.107268

*## Thank you very much for pointing this out. However, we don't believe that this conclusion from Fernando et al. could be linked to the biases seen at 35 km. We think TCOM biases are likely due to large dynamical variability, use of the measurements containing only positive values and limited spatial coverage in the tropics. Figure S2 clearly shows that R2 values for TCOM-CH4 in the tropics are almost constant (close to 0.5) from 25 to 50 km.*

>Line 242: "…in percentage terms biases can reach up to 100% near 40 km as changes in the small values can translate into much larger changes in relative differences."
In Figure 3, the shape of the percentage change looks quite similar to the HNO3 contribution to the N2O correction that is shown in Figure 1. The resemblance in shape looks even more pronounced for the SHmid case (Figures S3 and S9 in the supplementary file). To me, that

looks somewhat suspicious. Are you certain HNO3 is functioning as intended in the analysis? It looks like it is introducing a large percentage difference for N2O in the 2019-2021 evaluation period.

*## We have double-checked our code once again and the HNO3 proxy is correctly incorporated. The HNO3 importance being larger than other proxies most probably indicates that NOy (HNO3 being major contributor) species partitioning might be biased in TOMCAT.*

> Similar to CH4, a seasonal minimum occurs just after the break-up of Antarctic polar vortex (October), transporting N2O-depleted air to lower altitudes.
The seasonal minimum is not a consequence of the break-up of the polar vortex, it is the result of atmospheric descent within the polar vortex before it breaks up.

*## We agree; the paragraphs have been revised in the manuscript. However, the final vortex breakup only happens when there is major dynamical wave driving that suddenly strengthens stratospheric circulation. At the same time, this enhanced activity increases horizontal mixing as well (isentropic transport). So, major changes in the lower stratospheric CH4 happen only after polar vortex breakup. In contrast descending air masses inside the polar vortex, affect only small part of the stratosphere.*

>Line 262: "As the ACE-FTS retrieval algorithm uses multiple micro-windows, there may be a seasonal shift in averaging kernels causing fluctuations in the retrieved profiles."
ACE-FTS retrievals do not use averaging kernels. There is a seasonal variation in the spacing between tangent heights, and VMR profile variability could increase when tangent heights get very close together. When you get VMR values close to zero, it is normal to get negative values for an individual occultation. Note that excluding negative values and keeping only positive ones will actually introduce an artificial positive bias into averaged results.

*## We apologize for any confusion caused by our previous statement. We will add a sentence explaining the issue, as well as the possibility of positive biases in TCOM data because we only use positive values. We also plan to use both positive and negative values from ACE data during the construction of a new version of TCOM data.*

>Line 288: "The exact causes of unusually low CH4 values in S-MIPAS-CH4 and S-ACE-CH4 data files are unclear."
This is presumably another instance of atmospheric descent, with the descent signature in the data extending lower in altitude than in the model.

*## We will reiterate the role of the descending air masses once again.*

> Line 314: "…the latitude slice indicates significant variations between two".
…between the two.

*## Done*

---

## Referee Report (RR1)

The authors have answered all my comments to the previous version of the paper in a sufficient way. I only have a few technical comments to the revised version of the paper.

1. There are still several sentences (esp. in the newly added text) where e.g. 'the' is missing. I assume this will be handled during language editing.

2. l. 51:
   'Note that SPARC data uses earlier (v3.6) version of the ACE-FTS data' → 'Note that SPARC data use an earlier version (v3.6) of the ACE-FTS data.'

3. l. 52:
   'data contains' → 'data contain'

4. l. 215: 'We have added discussion in a revised manuscript'

   This sentence should be deleted, or do you refer to another manuscript?

5. l 316/317: 'Though exact causes of unusually low CH4 values in S-MIPAS-CH4 and S-ACE-CH4 data files are unclear, but might be associated with the downwards transport of CH4 poor air...'

   Unclear sentence, probably 'Though' should be deleted.

---

## Author Response (AR2)

**Referee #1**

**We thank the Dr Boone for insightful comments. Our replies are in the red coloured text.**

The authors have answered all my comments to the previous version of the paper in a sufficient way. I only have a few technical comments to the revised version of the paper.

1. There are still several sentences (esp. in the newly added text) where e.g. 'the' is missing. I assume this will be handled during language editing.

**Yes. We have tried to edit the text further but ESSD does also have dedicated team for style/grammar editing.**

2. l. 51: 'Note that SPARC data uses earlier (v3.6) version of the ACE-FTS data' → 'Note that SPARC data use an earlier version (v3.6) of the ACE-FTS data.'

**Done**

3. l. 52: 'data contains' → 'data contain'

**Done**

4. l. 215: 'We have added discussion in a revised manuscript' This sentence should be deleted, or do you refer to another manuscript?

**Done, we have removed the sentence.**

5. l 316/317: 'Though exact causes of unusually low CH4 values in S-MIPAS-CH4 and S-ACE-CH4 data files are unclear but might be associated with the downwards transport of CH4 poor air...' Unclear sentence, probably 'Though' should be deleted.

**Done, revised as "A possible explanation might be the absence of negative data points seen in ACE data (due to enhanced winter-time downward transport of CH4-poor air) which are excluded in the XGBoost training step."**

**Referee #2: (Chris Boone)**

The updated version of the manuscript shows improvement. I have no major concerns remaining, just some minor comments.

**We thank the Dr Boone for insightful comments. Our replies are in the red coloured text.**

>Line 245: Additionally, the onion peeling algorithm used for solar occultation measurements assumes observations at different tangent height are independent, hence retrieved profiles show larger fluctuations.
ACE-FTS analysis does not use onion peeling. The observational results will have 'random errors' contributing to the variability regardless of the analysis approach employed, for example from the impact of measurement noise in the analysis. The model data has no such contribution to the variability.
**Modified to say "some occultation instruments such as SAGE/ HALOE"**

> Line 289: As the ACE-FTS retrieval algorithm uses multiple micro-windows, a seasonal variation in vertical structure of the atmosphere alters spacing between tangent heights.
The spacing between tangent heights has no connection to the micro-windows. Spacing between tangent heights for ACE-FTS measurements exhibits a pseudo-seasonal variation as the occultation measurement geometry changes systematically over the course of a year.
**Thank you, we have removed the discussion about the micro-windows in the retrieval. Revised as "As the forward model used in the ACE-FTS retrieval algorithm needs spectra on fixed height levels, a seasonal variation in vertical structure of the atmosphere alters the spacing between tangent heights."**

>Line 166: We are aware that some tracers are correlated as all the variables are from a TOMCAT simulation (or forcing meteorology) …
Note that all the tracers used are also measured by ACE. It would be possible to separately evaluate each tracer, but that is presumably beyond the scope of this paper.
**Yes, we agree that it is beyond the scope for this paper.**

**Typos:**

>Line 51: Note that SPARC data uses earlier (v3.6) version of the ACE-FTS data
…uses an earlier…
**Done**

>Line 210: The time term (date) term…
'Term' is used twice
**Done**

>Line 258: Figures s5 and S6
One lower case s (s5) and one upper case (S6)
**Done**

>Line 284: a seasonal minima occurs
"minima" is plural.
**Done**

>Line 361: On the other hand, differences at 35 km in Figure 8 seem to be dominated by the QBO-induced meridional circulation patterns Baldwin et al. (2001), that are underestimated in TOMCAT. This sentence is not grammatically correct (perhaps …reported by Baldwin et al. or …detailed by Baldwin et al.).
**Modified as (e.g., Baldwin et al., 2001).**

**Referee #3**

On careful reading of the Conclusions, I find some confusion that may be just me or may be fixable with language:

**We thank the referee for his/her insightful comments and our replies are in the red coloured text.**

L388: 'constraint' is odd, the stratospheric profile are a key component of the X-CH4 that must be subtracted. I do not think of them as 'constraining'
**We think it is a constraint for the retrieval algorithm. We have modified the sentence to be "Also, accurate stratospheric CH4 profiles are a valuable constraint for the retrieval of tropospheric methane using satellite instruments."**

L390: likewise 'dynamical scheme' is strange. Numerics is also relevant. Why not just "dependent on the chemistry and transport processes in a particular model"
**Thank you for a useful suggestion. Revised the sentence as "Furthermore, although chemical models are able to simulate long-term profile data sets of these species, they are highly dependent on the representation of individual chemical and dynamical processes in a particular model."**

L398: "vary under". Do you mean they have different values in different location-times because of the regimes? or that they have different variability?
**Sorry for the confusion. We have revised and divided in to two sentences as "Because atmospheric concentrations of CH4 and N2O vary due to distinct dynamical and chemical processes in different regions, our approach involves dividing global measurements into five latitude-based categories. These categories include two for polar regions, two for mid-latitudes, and one for the tropics. We then proceed to derive regression parameters for each 1 km vertical grid spanning from 15 to 60 km within each of these latitude bins."**

L408: "the drift in Aura-MLS" This does not make it clear that the 'drift' is a bias or error, can you make it clear that it is an error?
**Added a reference and reworded as "confirms the abnormal drift in Aura-MLS v4.2 N2O data (as used in SPARC data set) especially at lower latitudes and altitudes (e.g., Livesey et al., 2021).**

L410: "near negligible trends …" The MLS N2O data show a clear positive trend in the upper strat that is greater than the surface trend, that is why the lifetime is dropping. What is meant here?
**We agree that MLS data shows positive trends in the upper stratosphere (Prather et al., 2023) but Minganti et al. (2022) show almost negligible trends in ACE-FTS data. We revised the sentences as:**

"A casual inspection of TCOM-CH4 and TCOM-N2O plots also suggests that despite increasing surface values there are near-negligible long-term trends in the upper stratosphere/lower mesosphere which is consistent with Minganti et al. (2022). On the other hand, Prather et al. (2022) analysed MLS V5 N2O data to showed positive trends (up to 15%) in the tropical upper stratospheric N2O, though they do not find NO production rising at the similar rates. A possible explanation would be variations in stratospheric/mesospheric loss processes, probably determined by changes in the stratospheric

circulation, is reducing the lifetime of these GHGs. We aim to analyse these discrepancies in a future study."